# Infant Food Security in New Zealand: A Multidimensional Index Developed from Cohort Data

**DOI:** 10.3390/ijerph16020283

**Published:** 2019-01-21

**Authors:** Deborah Schlichting, Ladan Hashemi, Cameron Grant

**Affiliations:** Department of Paediatrics: Child & Youth Health, The University of Auckland, Auckland 1142, New Zealand; l.hashemi@auckland.ac.nz (L.H.); cc.grant@auckland.ac.nz (C.G.)

**Keywords:** food security, paediatrics, cohort study, New Zealand, developed country

## Abstract

Food security (FS) during infancy is associated with lifelong outcomes. New Zealand is a developed economy that reports poor childhood nutrition-related health statistics, particularly among minority children, yet has no measure of FS applicable to infancy. The objective was to develop an FS index for New Zealand infants and examine its association with demographic covariates and health outcomes. Within a large (*n* = 6853) nationally representative cohort, variables describing infant food consumption, breastfeeding, and maternal food-related coping methods were collected from mothers during late infancy. An FS index was derived using confirmatory factor analysis. Associations were assessed by logistic regressions and described using odds ratios (OR) and ≥95% confidence intervals (CI). Fifteen percent of the cohort was highly FS, 43% tenuously food insecure (FIS), and 16% highly food insecure (FIS). Infants from minority ethnic groups had lower odds of being food secure, as did those born to the youngest mothers, mothers who smoked, or those who lived in low-income households. FIS infants had higher odds of morbidity. Interventions to improve infant FS should focus on improving dietary quality, and should give particular consideration to minority infants. We identified that FIS shows wide ethnic and socioeconomic inequity, and is associated with poorer health. The most important driving factors of FIS included poor quality weaning diets, as well as poverty and its proxies. Any interventions to improve infant FS should focus on increasing fruit and vegetable consumption to recommended intake levels, and should give particular consideration to minority infants.

## 1. Introduction

Food insecurity (FIS) refers to the lack of nutritious foods in sufficient quantities to maintain good health. FIS for infants and young children is gaining increased recognition in high-income countries [1] because of the prevalence of poor health outcomes associated with it, including obesity [2] and dental caries [3].

Suboptimal and inappropriate feeding associated with FIS is a major causes of undernutrition in young children [4]. Infants are particularly vulnerable to the adverse health effects of FIS because of their high nutritional requirements due to the demands of growth [5], and their dependence upon others for their nutritional requirements. Optimal nutrition during this period protects against morbidity and mortality, reduces the risk of chronic disease, and promotes better development overall [1].

Hallmarks of poor FS in a high-income country context are evident among New Zealand children, including the double burden of overweight/obesity and undernutrition [6,7], poor oral health [8,9], under-consumption of vegetables [10], high consumption of energy-dense nutrient-poor (EDNP) foods [11,12,13], micronutrient deficiencies [14], and hospital admissions for which nutritional deficiencies are contributory [15].

Indicators of FS for the New Zealand adult population [16] and households [17] have been developed, with the consideration of gender [16,18,19]; psychological distress [19]; households with children [11]; and ethnicity, specifically for Māori (New Zealand’s indigenous population) [20,21] and Pacific people [11,22]. In 2002, 51% of households with children described themselves as FS [9]. In 2004/05, 15% of the longitudinal Survey of Families, Income, and Employment population in New Zealand were FIS [18]. In 2017, the United Nations Children’s Emergency Fund (UNICEF) reported that 11% of New Zealand children aged <15 years live in food-insecure households [23]. Despite the importance of FS during infancy, there remains a paucity of evidence around this in the New Zealand context.

FS status is generally assessed at the household level by experience-based metrics or by food consumption data [24]. However, neither on its own is sufficient, particularly when the focus is on the individual [25]. New Zealand’s existing estimates of FIS may thus underestimate its actual prevalence because these studies used either experience scales or consumption data in their analysis.

To bridge this gap in the literature, following a review of the published literature and established tools, we develop a model of New Zealand infant FS, taking a multidimensional perspective to conceptualize it as being a function of both infant food consumption and FIS experiences of the mother. The approach of combining different dimensions of FS to develop a multidimensional measure of FS has been applied elsewhere [26,27]. A core finding of these efforts is that combining indicators can improve the measurement of FIS. Specifically, scores on food consumption, dietary diversity, and coping strategies are well correlated, suggesting that they all capture some element of the multidimensional notion of FS [27]. To validate the proposed FS index, we examine the relationship between demographic factors and infant FS, and between infant FS and infant health conditions.

## 2. Materials and Methods

Data on New Zealand infants were collected as part of GUiNZ, a nationally representative longitudinal cohort study of New Zealand children. Eleven percent of all children born in New Zealand during the recruitment period (2009–2010) was enrolled, and the study cohort was broadly generalizable in terms of ethnic diversity and markers of socio-economic status to all families having children in New Zealand today [28]. There were no exclusion criteria. Importantly, the study was designed to have adequate explanatory power to consider most outcomes for children who identify as Māori, Pacific, and Asian, as well as for New Zealand European children. No weighting was applied to participant enrollment. Details of GUiNZ methodology and study design are described elsewhere [28] and below. Ethical approval was obtained from the Ministry of Health Northern Y Regional Ethics Committee (NTY/08106/055). All enrolled women provided written informed consent.

All pregnant women living in a defined geographic region with an estimated delivery date between 25 September 2009 and 25 March 2010 were eligible for recruitment into the study. The geographical area chosen for recruitment was the region of New Zealand covered by the three contiguous District Health Board regions of Auckland, Counties-Manukau, and Waikato. This region was chosen for its ethnic and socioeconomic diversity. One-third of New Zealand births occur in this region [28]. Of the 10,315 referrals of pregnant women received, 6822 (66%) mothers consented to their child’s participation. The resulting cohort of 6853 children (live births) provides adequate statistical power to undertake complex analyses of interlinked developmental trajectories over time across the whole cohort, as well as within subgroups of children who identify as Māori, Pacific, and Asian (≥1000 children in each ethnicity). The demographic characteristics of the cohort at birth aligned with all births in New Zealand from 2007 to 2010. This study used data from 2010 when the infants were nine months old. We excluded infants for whom data on food consumption and/or breastfeeding and/or maternal coping was missing (*n* = 355), leaving a final sample of 6385 mothers and 6467 infants.

### 2.1. Methods

We treated infant FS as a latent construct, and used confirmatory factor analysis (CFA) to assess the extent to which the observed data agrees with theoretical concepts of FS. A four-factor model was adopted. The choice of the indicators has been informed by the literature, the aim of study, and the availability of data with sufficient coverage. In the proposed model, it was hypothesized that 19 items measure four constructs of food security—sentinel food group consumption (selected staple or nutrient-dense foods), EDNP consumption, maternal coping methods, and breastfeeding. Properties of the model were studied using CFA as implemented in AMOS 25.0 [29]. The analysis produces statistics reflecting overall fit of the observed data to the model, together with estimates of correlations between the items and factors. This modified model (15 items) for the overall sample showed adequate model fit, χ^2^ = 266.503, *p* < 0.001, root mean square error of approximation (RMSEA) = 0.03, Confirmatory Fit Index (CFI) = 0.94.

The observed variables were sentinel food group consumption (selected staple or nutrient-dense foods), EDNP consumption, maternal coping methods, and breastfeeding.

Maternal recall data on infant food consumption at an age of nine months provided information on types and frequencies of foods given to the child. With these data, we developed measures of daily consumption. Food groups were adapted from the WHO’s Indicators for Assessing Infant and Young Child Feeding practices (IYCF) [30]. Twelve individual food items were aggregated into 6 sentinel food groups, and 11 EDNP food and beverages were aggregated into 1 group (Table 1). The main IYCF indicators do not include a measure of EDNPs, but these were added as a separate indicator [30] because exposure to EDNP foods is of particular interest in the New Zealand context [13].

Breastfeeding measures are largely dependent upon available data. The most basic is a binary indicator of whether an infant was ever breastfed (IYCF indicator #9). More informative are indicators of duration of exclusive or predominant breastfeeding. Although IYCF suggests exclusive breastfeeding to an age of six months (IYCF indicator #2), our final choice of breastfeeding variable, as determined by CFA, was exclusive breastfeeding to an age of three months.

FS has both nutritional and non-nutritional pathways to well-being, and includes uncertain, insufficient, or unacceptable availability, access, or utilization of food. Measures that fail to capture disruptions in normal, socially-acceptable food acquisition practices and the adoption of more highly stigmatized to access food (e.g., dumpster-diving, theft, charitable food assistance) may understate the element of uncertainty that characterizes even mild FIS [31]. To incorporate risk and vulnerability, maternal coping methods were included in the model. GUiNZ surveyed mothers on their use of six different coping methods. Of these, four are directly relevant to FS—being forced to buy cheaper food, foregoing the consumption of fruit/ vegetables, receiving food assistance from a community organisation, and using food banks. In New Zealand, obtaining food from food banks is not regarded as ‘socially acceptable’ and is consequently an indicator of FIS [17].

After excluding subjects for whom data on food frequency, breastfeeding, and coping was incomplete or missing, the final sample size for the food security index was 6467 infants (see Appendix A Consort Diagram).

### 2.2. Statistical Analysis

Following previously established methods [32], the infant FS index was derived as a weighted sum of coping methods, food consumption, and breastfeeding (Table 2). In order to combine simplicity in assessing diet quality with the ability to differentiate healthy foods from unhealthy ones [33], punitive scoring is applied to unhealthy foods and use of coping methods. Positive points were awarded for the consumption of healthy foods and breastfeeding. Scoring was reversed and points deducted for consumption of unhealthy food and the use of coping methods. A zero-score was awarded to non-breastfeeding and not using coping methods. Each coping method used was scored −2 (else, 0) except ‘being forced to buy cheaper food’, which was scored −1 (else, 0). Exclusive breastfeeding to an age of three months was scored +2 for affirmative response (else, 0). Food consumption was measured as daily intake frequencies. Total daily sentinel food frequency was multiplied by +2, and total EDNP multiplied by −2. The range of scores was −20.29 to 29. For ease of discussion, a constant equal to the lowest value (−20.29) was added to all scores, shifting the range upward to 0–49.29. All statistical analyses were performed using SPSS version 25.

For index validation, we assessed the association between the FS index and commonly used socio-economic, demographic, and health outcome covariates using logistic regression. We set a binary variable for FIS at 1 (FIS) for infants who fall below the cut-point, and 0 above the cut-point.

## 3. Results

Infants’ dietary characteristics were assessed against WHO IYCF indicators (Table 3). Almost the entire cohort (96%) were breastfed at some point, 260 (2.5%) infants were never breastfed, 61% were exclusively breastfed for three months, and 78% for one month. By the age of 4 months, nearly 10% of the cohort had commenced complementary feeding, and by the age of 6 months, 70% was complementary feeding.

By IYCF guidelines, infants should be receiving food from ≥4 sentinel food groups (indicator #5) during complementary feeding, yet just 70% of the GUiNZ cohort achieve this. By World Food Programme (WFP) methods [34], “poor” FS scores indicate that households are falling short of consuming at least one starch food and one vegetable each day of the week. By this measure, 12% of the GUiNZ cohort would fall into the poor FS group. Around half, 54%, of the cohort was consuming iron-rich animal sourced food (haem iron). 

The majority of infants were eating fruit and vegetables at least daily (Table 4), although between 12% and 15% were not. Nearly 80% of the cohort reported consuming EDNP foods at least weekly, while 40% reported consuming these on a daily basis. Thirty percent had tried chocolate, 21% sweets, 20% crisps, 14% hot chips, and 5.4% soft drinks by the time they were nine months old. This is consistent with other New Zealand research, which found that 83% of children aged 5–14 years ate EDNPs at least weekly [11].

Half (54%) the cohort mothers reported using at least one coping method, and 18% used two or more methods (Table 5). The most commonly used coping method was buying cheaper food (*n* = 3237, 50%).

We found that infant FS was achieved by one-third of the cohort. The calculated food security index was normally distributed ($\bar{x}=25.52,\;s.d.=5.07$) with skewness of −0.004 and kurtosis of 0.397 (see Appendix A Histogram of Infant Food Security and Appendix A: Normal Q-Q Plot of Infant Food Security Index). We assessed normality using skewness and kurtosis of the distribution, because this method may be relatively correct in both small samples and large samples [35]. 

We assessed whether the index performed as expected against recognized FS covariates including socio-demographics (Table 6).

Consistent with the results from other countries [3,37], ethnic inequalities were pronounced. We found higher odds of FIS among ethnic minority infants including Māori (odds ratio (OR) 2.01: 95% confidence interval (CI) 1.75–2.29), Pacific (OR 2.98: 95% CI 2.54–3.49), and Asian infants (OR 1.27: 95% CI 1.09–1.46) when compared with all other infants. Poverty and its proxies gave infants higher odds of experiencing FIS, as did household factors such as crowding. The strongest factor was maternal smoking, which increased infant odds of FIS four-fold.

FIS status was significantly associated with poor health outcomes (Table 7). We found that FIS infants were 40% more likely to have a chest infection lasting more than a week compared with their FS peers. FIS infants were 30% more likely to have an ear infection, 35% more likely to see a doctor for gastroenteritis, and 25% more likely to experience sickness of any kind.

## 4. Discussion

While the choice of cut-off points in FS scales is often reported in the literature as seemingly arbitrary [27,32,38], the underlying justification for the choice of cut-off point is important. For instance, the cut-off point can be set by the political process to represent the minimum socially acceptable level of FS prevalence, so that governments would be concerned with managing the external drivers that push FS levels below the determined threshold [38]. Moreover, a slight change in the cut-off point can make a major difference in the magnitude of undernourishment, so that when the threshold shifts up, so too does the estimated prevalence of undernourishment. Accordingly, we set the cut-off point for infant FS at half a standard deviation above the mean. At the mean (25.5), the prevalence of infant FS was 50% (*n* = 3235). Shifting the cut-off point up by a half standard deviation to 28.03 reduced the prevalence of infant FS to 31% (*n* = 1994) (Table 8). Nearly half the cohort, whose scores fell within one-half standard deviation of the mean, were classified as tenuously FIS. At the extremes, 16% were highly or extremely FIS, while 15% were highly or extremely FS (Table 8).

This higher threshold was set for several reasons. First, as a high-income OECD country, New Zealand has the resources to ensure that all infants are highly FS, irrespective of their household FS. Second, it is unlikely that New Zealand, as a society, accepts infant FIS as necessary or inevitable. There is much public discourse around New Zealand’s problems with child poverty and hunger, suggesting that New Zealanders are not comfortable with any level of deprivation in childhood [39]. Third, this index deliberately seeks to establish a comprehensive picture of FS in the New Zealand context and includes threats to the stability of infant FS, the consumption of age-inappropriate foods (such as EDNPs) [13], and intake of good nutrition including breastfeeding. FS is not, we propose, a function of one factor alone, which means that it is possible to counter the downward pressure of factors such as poverty with the upward pressure of good nutrition.

We identified that infant FIS patterns are underpinned by low consumption of vegetables (76% consume ≤2 daily) and fruit (63% consume ≤2 daily), and by high exposure to EDNPs (12% consume ≥1 daily) (Table 5). Previous New Zealand research has found similar patterns in the national paediatric diet. A study using the 2002 New Zealand Child Nutrition Survey [10] found that EDNPs, specifically sugary foods and drinks, contributed to 20% of total energy intake of children’s diets. In 2008/2009, 40% of New Zealand 5–24 year olds reported consuming the recommendation for vegetable intake (≥3 servings/day) [10]. In a 2012–2014 study, [8] servings of fruit and vegetables in a cohort aged 5–17 years were below the recommended intake of both fruit and vegetable, and only 3% met the New Zealand recommendations for number of servings from the four main food groups. It is notable that this situation was already evident in 2002, when recommendations were made “to decrease intake of energy dense foods (particularly those containing saturated fats and sugars such as hot chips and sweet drinks) without compromising intake of essential nutrients” [11]. None of these studies included infants in their analyses. Our research specifically focuses on infants and demonstrates that suboptimal dietary patterns are already evident in infancy. Studies in other high-income contexts also reported that FIS is associated with lower intake of fruit/vegetables [31]. Our results are in line with these studies, confirming that FIS is associated with lower consumption of nutritious foods. 

Nutritious complementary feeding is critical to infant health. There is an established link between infant FIS and poor health outcomes [6,7,8,31,37,40,41,42]. Specifically, our results are consistent with research [43] that showed a relationship between FIS and infant respiratory infection.

A key observation from this research is that the incidence of infant FIS is high given the resource context of New Zealand. However, this is consistent with the age profile of poverty in New Zealand where the youngest are most at risk of poverty. In 2015, 14% of children aged 0–17 years lived in income poverty compared with those aged 18–25 (9.6%), those aged 18–65 (9.7%), and those over 65 (10.6%) [44].

From a policy perspective, the underlying elements of the index provide insight into the drivers of FIS. Increased intake of fruit/vegetables could readily increase FS rates. Of the infants ranked as tenuously FIS ($\bar{x}$ ± ½ s.d, *n* = 2475), 80% (*n* = 1975) could be shifted into the FS category ($\geq \bar{x}+½s.d.$) by consuming fruit/vegetables twice daily.

More difficult to shift are extremely FIS infants. For them, dietary quality is just one of several challenges to FS. The greatest burden of FIS lies with around 16% (*n* = 1020) of infants ($\leq \bar{x}-1s.d.$). These infants need to increase their scores by 8–28 points to shift into the range of FS, which requires modification in most, if not all, elements of the index.

Suggesting that New Zealand infants should consume more fruit/vegetables to improve their FS status ignores the difficulties that many households, most particularly the poor, face in accessing such food. Half of the mothers report having to buy cheaper food to pay for other necessities. Given the inverse relationship between diet cost and diet quality [45], many mothers may face difficulties in increasing the fruit/vegetables content of their baby’s diet.

Maternal coping is perhaps even more difficult to modify because it reflects degrees of hardship that transcend infant FS, and must necessarily be addressed by wider government policy.

Even though aggregate demand for food in New Zealand is relatively price inelastic [46], facilitating access to a better diet through price policies could still change consumption patterns. Low-income household and Māori demand for some sentinel foods is elastic [47]. Households in income quintile 1 (poorest) have an own-price-elasticity of vegetables of −1.09. The New Zealand sales tax (GST) is levied at 15%, which would equate to a 16.4% reduction in vegetable consumption for this group. From a policy perspective, a consideration of the role that GST plays in diet quality may be warranted, particularly given WHO advice that, to achieve FS, increased access to foods of good nutritional quality should be ensured in all local markets at an affordable price all year round [1]. Moreover, New Zealand’s closest neighbor by geography and socio-economics, Australia, does not levy sales tax on many sentinel foods including fruit, vegetables, fish, and meat [48].

Breastfeeding is important for infant FS. Rates and duration of exclusive breastfeeding are low in this cohort compared with international guidelines. Exclusive breastfeeding rates fall from 61% at an age of 3 months to 24% at an age of 6 months. Breastfeeding continuity may be obstructed by parental leave legislation, which, in 2009, gave mothers 14 weeks’ paid leave. By 2018, parental leave has increased to 22 weeks, with a further increase to 26 weeks is scheduled for 2020 [49]. This may increase national breastfeeding rates, which would help improve infant FS.

The strengths of our study include that it is, to our knowledge, the only infant food security index for New Zealand. The index is further unique in its multidimensional structure that allows for a targeted focus on infants. Some limitations of our study are worth noting. Our primary source of infant dietary information was from maternal recall data, which may not be the best representation of an infant’s usual intake because of the variation in daily intake. Second, because this is an observational study, residual confounding cannot be ruled out completely. Third, there is the potential that characteristics of the pregnant women who registered, but whose children were not enrolled into the cohort, differ from those of the children who formed the cohort. As no information was collected on those who registered but were not enrolled, we are unable to comment further on this. We were, however, able to confirm that characteristics of the cohort at birth closely aligned with all births in New Zealand in 2007–2010 [28].

## 5. Conclusions

We identified that FIS, to some extent, affects around 72% of New Zealand infants, shows wide ethnic and socioeconomic inequity, and is associated with poorer health. The most important driving factors of FIS included poor quality weaning diets, as well as poverty and its proxies. Any interventions to improve infant FS should focus on increasing fruit and vegetable consumption to recommended intake levels, and give particular consideration to Māori and Pacific infants. Within this nationally representative cohort, we found 16% of infants were highly or extremely FIS, and 43% were tenuously FIS. This is consistent with estimates of New Zealand household food insecurity in 2001 [18], and Canadian [50] and U.S. [51] findings.

The large inequities that we found in infant FS by ethnicity and deprivation signal a need to focus specifically on Māori and Pacific infants, and more socioeconomically deprived communities with any interventions to address infant FIS in New Zealand.

We found that FIS during the period of complementary feeding, particularly because of low consumption of fruit and vegetables and high and early consumption of EDNPs, is a risk for many infants, most particularly for Māori and Pacific infants. FIS and its consequences are a problem for New Zealand infants and more work needs to be done on understanding and addressing it.

## Figures and Tables

**Table 1 ijerph-16-00283-t001:** Food items and food groups created from the Growing Up in New Zealand study as measures of the WHO Infant and Young Children Feeding Indicator (IYCF). EDNP—energy-dense nutrient-poor.

IYCF Food Groups	GUiNZ ^a^ Food Groups	GUiNZ Food Items
Grains, roots, and tubers	Grains	Baby rice
		Baby cereal
		Other cereal
		Bread/toast
		Rusks
Legumes and nuts	Legumes and nuts	Nuts, peanut butter ^b^
		Soy foods, tofu, soy dessert ^b^
Flesh foods (meat, fish, poultry, & liver/organ)	Meat and chicken	Meat, chicken, meat dishes
	Fish and shellfish	Fish (fresh or canned) ^b^
		Shellfish ^b^
Eggs	Eggs	Eggs ^b^
Vitamin-A rich fruits & vegetables	Vegetables	Vegetables (raw or cooked)
Other fruits and vegetables	Fruit	Fruit (fresh and canned)
Energy-dense/nutrient-poor foods	EDNP	Biscuits
		Milk & rice puddings, yoghurt, custards ^b^
		Sweets
		Chocolate
		Hot chips
		Crisps
		Fruit juice
		Herb drinks ^b^
		Tea ^b^
		Coffee ^b^
		Soft drinks

^a^ Growing Up in New Zealand. ^b^ excluded.

**Table 2 ijerph-16-00283-t002:** Food security index components, weights, scores, and ranges.

Component		Weight	Minimum	Maximum
Coping	Being forced to buy cheaper food	Y = −1	−1	0
		N = 0		
	Going without fruit/vegetables	Y = −2	−2	0
		N = 0		
	Help from charity	Y = −2	−2	0
		N = 0		
	Use foodbank	Y = −2	−2	0
		N = 0		
Breastfeeding	Exclusive breastfeeding to 3 months	Y = 2	0	2
		N = 0		
Sentinel foods	Daily consumption of sentinel foods	Q × 2 *	0	27
EDNP	Daily consumption of 6 EDNP foods	Q × −2 **	−13.29	0
			Min: −20.29	Max: 29
	Add constant of 20.29		Min: 0	Max: 49.29

* Q × 2 is daily consumption frequency (Q) multiplied by 2. ** Q × −2 is daily consumption frequency (Q) multiplied by −2.

**Table 3 ijerph-16-00283-t003:** Prevalence of Infant and Young Child Feeding Indicators (IYCF) in the Growing Up in New Zealand study.

IYCF Indicators		GUiNZ Data at Age 9 Months (*n*, %)
#2: Exclusive breastfeeding to age 6 months		1545 (24%)
#4 Introduction of solids at age 6–8 months		4526 (70%)
#5 Minimum dietary diversity (≥3 food groups)		4526 (70%)
#6 Minimum food frequency (≥4/day)		6078 (94%)
#7 Minimum acceptable diet		4526 (70%)
#8 Consumption of iron-rich foods (haem-iron) ^a^		3492 (54%)
#9 Ever breastfed		6208 (96%)
**Additional Indicators**		
Exclusive breastfeeding to age 3 months		3944 (61%)
Early introduction of solids (≤ age 4 months)		
Baby rice		1895 (29%)
Fruit		1320 (20%)
Vegetables		1178 (18%)
Chocolate		159 (3%)
Daily or greater consumption of EDNP foods		2586 (40%)
Weekly or greater consumption of EDNP foods		5108 (79%)
Maternal food related coping methods, any		3492 (54%)
Maternal food related coping methods, more than one		1164 (18%)
Coping Methods	Cheaper food	3237 (50%)
	No fruit/veg	811 (13%)
	Charity	347 (5%)
	Food banks	851 (13%)

^a^ We include only haem-iron, but IYCF includes non-animal iron-fortified foods in this measure.

**Table 4 ijerph-16-00283-t004:** Frequency of food group consumption at an age of 9 months for infants enrolled in the Growing Up in New Zealand study.

	Food Groups, *n* (%)	Never	>0 – <1/d	>1 – <2/d	>2 – ≤3/d	≥3/d
Sentinel Foods	Vegetables	255 (4%)	540 (8%)	3559 (55%)	1910 (30%)	203 (3%)
	Fruit	216 (3%)	760 (12%)	3103 (48%)	1816 (28%)	572 (9%)
	Grains	248 (4%)	529 (8%)	667 (10%)	2856 (44%)	2167 (34%)
	Meat/chicken	863 (13%)	2188 (34%)	2900 (45%)	461 (7%)	55 (1%)
	Fish	4129 (64%)	2129 (33%)	165 (3%)	38 (1%)	6 (0.1%)
	Legumes	5331 (82%)	945 (14%)	153 (2%)	29 (0.45%)	9 (0.14%)
EDNP Foods	EDNP foods	1384 (21%)	2491 (39%)	1912 (30%)	515 (8%)	165 (2.5%)
	Sugar Sweetened Beverages	5001 (77%)	857 (13)	440 (6.8%)	128 (2%)	41 (0.6%)

**Table 5 ijerph-16-00283-t005:** Food security components by status.

Component			Food Security Status	
			Food Secure	Food Insecure
Coping	Being forced to buy cheaper food	Yes	1238 (23%)	2483 (77%)
		No	754 38%)	1992 (62%)
	Going without fruit/vegetables	Yes	56 (7%)	755 (93%)
		No	1936 (34%)	3720 (66%)
	Help from charity	Yes	17 (5%)	332 (95%)
		No	1975 (32%)	4143 (68%)
	Use foodbank	Yes	53 (6%)	798 (94%)
		No	1939 (35%)	3677 (65%)
Breastfeeding to 3 months		Yes	1552 (39%)	2380 (61%)
		No	440 (17%)	2095 (83%)
Sentinel Foods	Daily consumption frequency, mean (s.d.)		7.59 (1.61)	4.39 (1.42)
EDNP	Daily consumption frequency, mean (s.d.)		0.10 (0.37)	0.36 (0.78)

**Table 6 ijerph-16-00283-t006:** Adjusted odds ratios (OR) for the association of demographic and socio-economic characteristics with infant food security status.

	*n* (%) ^g^	Food Secure Group	Food Insecure Group ^e^	*p* Value	OR (95% CI)	*p* Value
		*n* (%)	*n* (%)			
Mother forced to put up with feeling cold	6467			<0.001		
No (ref)	5279 (82)	1764 (89)	3515 (79)		1.00	
Yes	1188 (18)	228 (11)	960 (21)		2.11 (1.81–2.49)	<0.05
Mother forced to wear shoes with holes	6467			<0.001		
No (ref)	5684 (88)	1837 (92)	3847 (86)		1.00	
Yes	783 (12)	155 (8)	628 (14)		1.94 (1.60–2.36)	<0.05
Deprivation Index ^a,b^	6382			<0.001		
≤3: Low (ref)	1671 (26)	705 (36)	966 (22)		1.00	
4–7: Medium	2343 (37)	813 (41)	1530 (35)		1.37 (1.21–1.56)	<0.05
8–10: High	2368 (37)	457 (23)	1911 (43)		3.05 (2.65–3.51)	<0.05
Obtained prescription for baby but didn’t collect ≥1 items from the chemist because you could not afford	6382			<0.001		
No (ref)	6172 (97)	1945 (98)	4227 (96)		1.00	
Yes	210 (3)	30 (2)	180 (4)		2.76 (1.86–4.07)	<0.05
Any difficulty paying for medical care or medicines that your baby needed?	6382			<0.001		
No (ref)	6157 (96)	1945 (98)	4212 (96)		1.00	
Yes	225 (4)	30 (2)	195 (4)		3.00 (2.03–4.42)	<0.05
Ethnicity ^c^				<0.001		
Māori	6467					
Yes	1548 (24)	317 (16)	1231 (28)		2.01 (1.75–2.29)	<0.05
No (ref)	4919 (76)	1675 (84)	3244 (72)		1.00	
Pacific	6382					
Yes	1364 (21)	209 (11)	1155 (26)	<0.001	2.98 (2.54–3.49)	<0.05
No (ref)	5018 (79)	1766 (89)	3252 (74)		1.00	
Asian	6382					
Yes	1085 (17)	292 (15)	793 (18)	<0.001	1.27 (1.09–1.46)	<0.05
No (ref)	5297 (83)	1683 (85)	3614 (82)		1.00	
MELAA ^d^	6382			<0.05		
Yes	180 (3)	62 (3)	118 (3)		0.84 (0.62–1.15)	NS
No (ref)	6202 (97)	1913 (97)	4289 (97)		1.00	
European	6382			<0.001		
Yes	4424 (69)	1618 (82)	2806 (64)		0.39 (0.34–0.44)	<0.05
No (ref)	1958 (31)	357 (18)	1601 (36)		1.00	
Rurality	6382			<0.05		
No (ref)	5905 (93)	1822 (92)	4083 (93)		1.00	
Yes	477 (7)	153 (8)	324 (7)		1.05 (0.86–1.29)	NS
Mother smoker	6467			<0.001		
No (ref)	5556 (86)	1890 (95)	3666 (82)		1.00	
Yes	911 (14)	102 (5)	809 (18)		4.09 (3.30–5.06)	<0.05
Mother age group at pregnancy	6467			<0.001		
<20	292 (5)	33 (2)	259 (6)		4.81 (3.33–6.96)	<0.05
20–29	2482 (38)	554 (28)	1928 (43)		2.13 (1.90–2.40)	<0.05
>30 (ref)	3693 (57)	1405 (71)	2288 (51)		1.00	
Number of people aged <18 year in house	6379			<0.001		
One or Two (ref)	4562 (72)	1576 (80)	2986 (68)		1.00	
Three	1086 (17)	292 (15)	794 (18)		1.43 (1.23–1.66)	<0.05
>Four	731 (11)	106 (5)	625 (14)		3.11 (2.51–3.86)	<0.05
Household crowding	6381			<0.001		
<1: low (ref)	354 (6)	133 (7)	221 (5)		1.00	
≥1 to <2: medium	4632 (73)	1635 (83)	2997 (68)		1.10 (0.88–1.37)	NS
≥2: high	1395 (22)	207 (10)	1188 (27)		3.45 (2.66–4.48)	<0.05

^a^ New Zealand Deprivation Index (NZDep) 2006 [36]. ^b^ NZDep is used as a proxy for income because household income data in Growing Up in New Zealand are not reliable. ^c^ Infant’s ethnicity as described by parents. ^d^ MELAA: Middle East, Latin America, Asia. ^e^ Food insecure includes those children classified as tenuously food secure. ^f^ Values are adjusted odds ratios from multivariable logistic regression models (95% confidence interval (CI)). The multivariable models were adjusted for material hardship (mother forced to put up with feeling cold, and mother forced to wear shoes with holes), neighborhood deprivation, inability to have prescription for baby filled, difficulty paying for baby’s medication, baby’s ethnicity, rurality, maternal smoking, maternal age, number of children in household, and household crowding. ^g^ If *n* < 6467, incomplete and missing data have reduced the sample size. NS—not significant.

**Table 7 ijerph-16-00283-t007:** Adjusted odds ratios^a^ for the association of food security status with health outcome in nine month old infants.

	*n* (%)	Food Secure Group	Food Insecure Group	*p*-Value (χ^2^)	OR (95% CI)	*p*-Value
		*n* (%)	*n* (%)			
*Experiencing any sickness*	6467			<0.005		<0.05
*Never*	4420 (68)	1291 (65)	3129 (70)		1.00	
*At least once*	2047 (32)	701 (35)	1346 (30)		1.26 (1.13–1.41)	
*Seeing a doctor for any sickness*	6467			<0.03		<0.05
*Never*	4740 (73)	1425 (72)	3315 (74)		1.00	
*At least once*	1727 (27)	567 (28)	1160 (26)		1.14 (1.01–1.28)	
*Experiencing chest infection, wheezing, bronchiolitis, bronchitis, asthma lasting more than one week*	6467			<0.001		<.001
*Never*	5244 (81)	1681 (84)	3563 (80)		1.00	
*At least once*	1223 (19)	311 (16)	912 (20)		1.38 (1.20–1.59)	
*Seeing a doctor for chest infection lasting more than 1 week*	6467			<0.001		
*Never*	4806 (74)	1543 (77)	3263 (73)		1.00	
*At least once*	1661 (26)	449 (23)	1212 (27)		1.28 (1.13–1.44)	<0.05
*Experiencing ear infection*	6467			<0.001		<0.05
*Never*	5005 (77)	1601 (80)	3404 (76)		1.00	
*At least once*	1462 (23)	391 (20)	1071 (24)		1.29 (1.13–1.47)	
*Seeing a doctor for ear infection*	6467			<0.001		
*Never*	5035 (78)	1610 (81)	3425 (77)		1.00	
*At least once*	1432 (22)	382 (19)	1050 (23)		1.29 (1.13–1.47)	<0.05
*Experiencing cough lasting more than one week*	6467			<0.001		<0.05
*Never*	3548 (55)	1156 (58)	2392 (53)		1.00	
					1.20 (1.08–1.34)	
*At least once*	2919 (45)	836 (42)	2083 (47)			
*Seeing a doctor for cough lasting more than one week*	6476			<0.001		
					1.00	
*Never*	3971 (61)	1301 (65)	2670 (60)		1.27 (1.14–1.42)	<0.05
*At least once*	2496 (39)	691 (35)	1805 (40)			
*Experiencing gastroenteritis*	6467			<0.003		
					1.00	
*Never*	5063 (78)	1605 (81)	3458 (77)		1.22 (1.07–1.39)	<.003
*At least once*	1404 (22)	387 (19)	1017 (23)			
*Seeing a doctor for gastroenteritis*	6467			<0.001	1.00	
*Never*	3971 (61)	1301 (65)	2670 (60)			
					1.35 (1.16–1.58)	
*At least once*	2496 (39)	691 (35)	1805 (40)			<0.05
*Experiencing eczema*	6467			<0.004		
*Never*	4318 (67)	1280 (64)	3038 (68)		1.00	
*At least once*	2149 (33)	712 (36)	1437 (32)		0.85 (0.76–0.095)	<0.004
*Seeing a doctor for eczema*	6467			0.3 (NS)		
*Never*	4789 (74)	1460 (73)	3329 (74)		1.00	
*At least once*	1668 (26)	532 (27)	1146 (26)		0.94 (0.83–1.06)	0.3(NS)
*Experiencing skin infection*	6467			0.2 (NS)		0.2 (NS)
*Never*	5931 (92)	1815 (91)	4116 (92)		1.00	
*At least once*	536 (8)	177 (9)	359 (8)		0.89 (0.74–1.07)	
*Seeing a doctor for skin infection*	6467			0.2 (NS)		0.2 (NS)
*Never*	5994 (93)	1834 (92)	4160 (93)		1.00	
*At least once*	473 (7)	158 (8)	315 (7)		0.87 (0.72–1.07)	

^a^ Values are adjusted odds ratios from multivariable logistic regression models (95% CIs). The multivariable models were adjusted for material hardship (mother forced to put up with feeling cold, and mother forced to wear shoes with holes), neighbourhood deprivation, inability to have prescription for baby filled, difficulty paying for baby’s medication, baby’s ethnicity, rurality, maternal smoking, maternal age, number of children in household, and household crowding. NS—not significant.

**Table 8 ijerph-16-00283-t008:** Food security cut points.

Cut Point (Score)	Definition	Prevalence, *n* (%)
<−2 s.d. (15.39)	Extremely food insecure	162 (2.5%)
−2 s.d. ≤ • <−1 s.d. (20.45)	Highly food insecure	858 (13.3%)
−1 s.d ≤ • < −0.5 s.d. (23)	Moderately food insecure	876 (13.5%)
−0.5 s.d. ≤ • < +0.5 s.d. (28.03)	Tenuously food insecure	2768 (42.8%)
+0.5 s.d. ≤ • < +1 s.d. (30.87)	Moderately food secure	817 (12.6%)
+1 s.d. ≤ • < +2 s.d. (35.65)	Highly food secure	876 (13.5%)
≥ +2 s.d.	Extremely food secure	110 (1.7%)

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
