# Peer review of "Infant Food Security in New Zealand: A Multidimensional Index Developed from Cohort Data"

_ijerph, 2019, doi:10.3390/ijerph16020283_

Round 1

Reviewer 1 Report

Food security is for sure a key concept for public health nutrition. Though, since food insecurity has determinants of various dimensions, its identification is a very complex issue. Most of the scales and indicators used to identify food insecurity turn to one of these dimensions, resulting in their low validity. In this context, the development of a multidimensional index to identify food insecurity appears to be very useful. However, several questions regarding the methodology used and the presentation of the results require further work. While important points of the methodology used are not clear, the volume of results presented seems to be excessive in some analyzes and insufficient in others. As a result, the discussion shows disconnection with the results presented. These issues are detailed below.

L70-4: Once the used cohort is mentioned as “nationally representative”, further details about sampling procedure are necessary. Furthermore, in the following paragraph it is mentioned that pregnant women living in a “defined geographic region” were eligible. If the cohort is truly nationally representative, this second part must be better explained.

L119-24: The definition of the weight for each variable is unclear. As this has a lot of impact in results, a rational for this must be exposed.

L126: Is data from GUiNZ self-weighted? If not, were the sampling (and maybe post-stratification) weights considered? What about non-response, how was this handled?

L139: Table 3 information is already presented on text. Suggest removing the table.

L151-3: Table 6 and 7 are based in a single regression model adjusted by all the variables or by several models?  As most of these variables are known to be highly correlated, what’s is gained with all the independent association values?

L177-8: I suggest including disaggregated coping information on Table 4. Current information may remain as it is, but information on each coping method could be also presented.

L179-0: Reference manager error.

L226-33: The fraction of the score attributed to each component is not presented in the results. Thus, most of the discussion becomes very disconnected from the results. I strongly suggest having figure 1 replaced by a table containing all levels of FS (and FIS) (as columns) and the components of the index (as lines). This would allow the reader to understand how each component of the index varied across the FS (or FIS) levels.

L241-48: Price policies are just one way to increase access, and if the demand is admittedly inelastic, why the authors focused on this kind of policy?   

Author Response

Reviewer 1

Comments and Suggestions for Authors 

Food security is for sure a key concept for public health nutrition. Though, since food insecurity has determinants of various dimensions, its identification is a very complex issue. Most of the scales and indicators used to identify food insecurity turn to one of these dimensions, resulting in their low validity. In this context, the development of a multidimensional index to identify food insecurity appears to be very useful.

Response:  Thank you very much for these positive comments regarding the potential utility of our multidimensional food security index.

However, several questions regarding the methodology used and the presentation of the results require further work. While important points of the methodology used are not clear, the volume of results presented seems to be excessive in some analyzes and insufficient in others. As a result, the discussion shows disconnection with the results presented. These issues are detailed below. 

Response:  Thank you. We will address these issues point by point.

L70-4: Once the used cohort is mentioned as “nationally representative”, further details about sampling procedure are necessary. Furthermore, in the following paragraph it is mentioned that pregnant women living in a “defined geographic region” were eligible. If the cohort is truly nationally representative, this second part must be better explained.

Response:  The following statements were added to address the national representativeness of the sample and to describe why this regional approach was used for study recruitment:

L71-77: “Eleven percent of all children born in New Zealand during the recruitment period (2009-10) was enrolled, and the study cohort is broadly generalisable in terms of ethnic diversity and markers of socio-economic status to all families having children in New Zealand today.[29]  There were no exclusion criteria.  Importantly, the study was designed to have adequate explanatory power to consider most outcomes for children who identify as Māori, Pacific, and Asian as well as for New Zealand European children.  No weighting was applied to participant enrolment. 

An additional reference has been included to support this:

Morton, S.M.B.; Atatoa Carr, P.E.; Grant, C.C.; Robinson, E.M.; Bandara, D.K.; Bird, A.; Ivory, V.C.; Kingi, T.K.R.; Liang, R.; Marks, E.J., et al. Cohort Profile: Growing Up in New Zealand. International Journal of Epidemiology 2013, 42, 65-75, doi:10.1093/ije/dyr206.

L82-85:  “The geographical area chosen for recruitment was the region of New Zealand covered by the three contiguous District Health Board regions of Auckland, Counties-Manukau, and Waikato. This region was chosen for its ethnic and socioeconomic diversity. One third of New Zealand births occur in this region.[29]

L119-24: The definition of the weight for each variable is unclear. As this has a lot of impact in results, a rational for this must be exposed.

Response: The following statements were added (Ln129-134): “In order to combine simplicity in assessing diet quality with the ability to differentiate healthy foods from unhealthy ones[33], punitive scoring is applied to unhealthy foods and use of coping methods. Positive points were awarded for the consumption of healthy foods and breastfeeding. Scoring was reversed and points deducted for consumption of unhealthy food and the use of coping methods. A zero-score was awarded to non-breastfeeding and not using coping methods.”

An additional reference was included to support this [33]: Fung, T.T.; Isanaka, S.; Hu, F.B.; Willett, W.C. International food group–based diet quality and risk of coronary heart disease in men and women. The American Journal of Clinical Nutrition 2018, 107, 120-129, doi:10.1093/ajcn/nqx015.

L126: Is data from GUiNZ self-weighted? If not, were the sampling (and maybe post-stratification) weights considered? What about non-response, how was this handled?

Response: No, data from GUiNZ was not self-weighted. No weighting was applied when the pregnant women were enrolled into the study. Eligibility for enrolment was based only upon residence in a geographically defined region of New Zealand.

The following statements was added (Ln76-77): “No weighting was applied to participant enrolment.”

Responses with missing or incomplete data relating to the food security index were excluded. We provide the following Consort Diagram to show this (this consort diagram is also included as a .pdf file):

L139: Table 3 information is already presented on text. Suggest removing the table.

Response:  Thank you, in response to Reviewer 2’s comments below, we have moved the text from Ln128-150 and Table 3 to the Discussion section, and these now appear in Ln196-217.

L151-3: Table 6 and 7 are based in a single regression model adjusted by all the variables or by several models?  As most of these variables are known to be highly correlated, what’s is gained with all the independent association values?

Response:  Thank you. Tables 6 and 7 report adjusted odds ratios from multivariable logistic regression models, controlling for socio-demographic factors.  We have added a footnote row to both Tables 6 and 7 to clarify this:

Values are adjusted odds ratios from multivariable logistic regression models (95% CIs).  The multivariable models were adjusted for material hardship (mother forced to put up with feeling cold, and mother forced to wear shoes with holes), neighbourhood deprivation, inability to have prescription for baby filled, difficulty paying for baby’s medication, baby’s ethnicity, rurality, maternal smoking, maternal age, number of children in household, and household crowding.

L177-8: I suggest including disaggregated coping information on Table 4. Current information may remain as it is, but information on each coping method could be also presented.

Response: Thank you. Done as requested. The following rows have been added to the Table which is now Table 3 (please see attached file for the table).

L179-0: Reference manager error.

Response: Thank you. This has now been corrected (Ln170).  The reference was a cross-reference to Figure 1, which has been removed in response to this reviewer’s suggestion to do so. 

L226-33: The fraction of the score attributed to each component is not presented in the results. Thus, most of the discussion becomes very disconnected from the results. I strongly suggest having figure 1 replaced by a table containing all levels of FS (and FIS) (as columns) and the components of the index (as lines). This would allow the reader to understand how each component of the index varied across the FS (or FIS) levels.

Response: Thank you for this suggestion. Figure 1 has been removed, and replaced with Table 5 (Ln 176) (please see the attached file which includes this table).

L241-48: Price policies are just one way to increase access, and if the demand is admittedly inelastic, why the authors focused on this kind of policy?   

Response: We have rephrased the statement from the original “Even though demand for food in New Zealand is relatively price inelastic,[44] facilitating access to a better diet through price policies could still change consumption patterns.  Low-income households and Māori show higher price-elasticity of demand” to: “Even though aggregate demand for food in New Zealand is relatively price inelastic,[45] facilitating access to a better diet through price policies could still change consumption patterns.  Low-income households and Māori demand for some sentinel foods is elastic”.

We have also added this statement to support the focus on price policy:

Moreover, New Zealand’s closest neighbour by geography and socio-economics, Australia, does not levy sales tax on many sentinel foods including fruit, vegetables, fish, and meat. [47]

And we have provided an additional reference to support this statement : Australian Tax Office. GST-free food. Availabe online: https://www.ato.gov.au/Business/GST/In-detail/Your-industry/Food/GST-and-food/?anchor=GSTfreefood#GSTfreefood (accessed on 26/11/2018).

Reviewer 2 Report

This is a interesting study addressing a very important topic. The paper is clear and well-written and the study utilises well established cohorts and suitable data collection and statistical methods.

Introduction is well-written and clearly explains the background for the study.

The data were collected in 2010 and thus may be considered a bit old. Could the authors comment this – does the data reflect also the current situation in NZ  or has something changed?

The final sample was 6467 infants from 10315 registered pregnancies. Is there a possibility of resulting bias in the findings? Perhaps this possibility should be mentioned as a study limitation.

Was there any information available on maternal education?

There are some very important points mentioned in the methods section that I would rather see in the discussion, such as the choice of cut-offs and how it affects the findings in lines 128-135 and lines 141-150.

Was there any confounders available that could have been taken in account when analyzing the findings presented in Table 7? F ex, birth characteristics or socioeconomic status?

Conclusions, line 273 the authors state that FIS during the period of complementary feeding is a risk for many infants. Could this be specified?

Author Response

Reviewer 2

Comments and Suggestions for Authors

This is an interesting study addressing a very important topic. The paper is clear and well-written and the study utilises well established cohorts and suitable data collection and statistical methods.

Introduction is well-written and clearly explains the background for the study.

The data were collected in 2010 and thus may be considered a bit old. Could the authors comment this – does the data reflect also the current situation in NZ or has something changed?

Response: Thank you for this observation. The cohort children were born in 2009 and 2010 and so the data used to create the infant food security index was collected in 2010 and 2011.  We intend to use this food security index, as calculated for the cohort when they were infants, to assess outcomes later in life that are expected to be influenced by early life food security.  Thus this food security index will remain relevant to this cohort.  The ethnic and socioeconomic diversity of the cohort remains comparable to that of current annual birth cohorts in New Zealand and thus the index remains relevant to the current situation in New Zealand.

The final sample was 6467 infants from 10315 registered pregnancies. Is there a possibility of resulting bias in the findings? Perhaps this possibility should be mentioned as a study limitation.

Response:  The intention of the recruitment strategy used to create this cohort was to enrol a sample that was generalizable to the national birth cohort rather than to the cohort of children born in the study region.  Following enrolment of the cohort, it was demonstrated that the characteristics of the cohort at birth closely aligned with the cohort closely aligned to all births in New Zealand in 2007-2010.  The reviewer is correct to pose the question of whether there is the potential that of the 10315 registered pregnancies there were differences between those who were enrolled versus those not enrolled.  However, as no data were collected on the registered but not enrolled pregnancies no comparisons can be made.

The following test has been added (Ln278-283):

Third, there is the potential that characteristics of the pregnant women who registered, but whose children were not enrolled into the cohort, differ from those of the children who formed the cohort. As no information was collected on those who registered but were not enrolled we are unable to comment further on this.  We were, however, able to confirm that characteristics of the cohort at birth closely aligned with the cohort closely aligned to all births in New Zealand in 2007-2010.[29]

Was there any information available on maternal education?

Response:  Information on maternal education is available for the Growing Up in New Zealand cohort.  We chose not include variables describing maternal education in our analyses because these caused multicollinearity problems when included alongside other socio-demographic variables such as neighbourhood deprivation.

There are some very important points mentioned in the methods section that I would rather see in the discussion, such as the choice of cut-offs and how it affects the findings in lines 128-135 and lines 141-150.

Response:  Thank you. Done as requested.  The material that was previously in the methods section (Ln128-150 and Table 3) has been moved to the discussion section, and now appears in Ln 196-217 and Table 8.  No alteration has been made to the text.

Was there any confounders available that could have been taken in account when analyzing the findings presented in Table 7? F ex, birth characteristics or socioeconomic status?

Response:  Yes, the same confounding variables that were used in the analysis reported in Table 6 were used in the analyses reported in Table 7.  We have changed the titles of both Table 6 and 7 to include “Adjusted Odds Ratios of”.  The title for Table 6 is now “Adjusted Odds Ratios for the association of demographic and socio-economic characteristics with infant food security status.”   The title of Table 7 is now “Adjusted Odds Ratios for the association of food security status with health outcome in nine month old infants

In addition, we have added a footnote row to table 7: Values are adjusted odds ratios from multivariable logistic regression models (95% CIs).  The multivariable models were adjusted for material hardship (mother forced to put up with feeling cold, and mother forced to wear shoes with holes), neighbourhood deprivation, inability to have prescription for baby filled, difficulty paying for baby’s medication, baby’s ethnicity, rurality, maternal smoking, maternal age, number of children in household, and household crowding.

Conclusions, line 273 the authors state that FIS during the period of complementary feeding is a risk for many infants. Could this be specified?

Response: We have added a sentence into the paragraph: “We found that FIS during the period of complementary feeding, particularly due to low consumption of fruit and vegetables and high and early consumption of EDNPs, is a risk for many infants, most particularly for Māori and Pacific infants.”

Reviewer 3 Report

This is an interesting study. Authors want to use a integrated index to assess the risk of infant food security. This simple index might be useful or implication for child health care. However, the following questions should be notes.

1. In the part of methods, the concept of index establishment should be presented in detail. How many subjects were used for setting up index and how many subjects for verification?

2. Authors mentioned the CFA was used for establishment of index. However, there was nothing to say about this CFA. If CFA was used, the theoretical concept about this index should be reported and the procedure to test it by CFA was also reported.

3. The calculated index FIS was normal distribution? or what kind of distribution for it? this problem is unclear.

4. for table 2, the weights were unclear. what is criteria for the weights? Some statement might be necessary. And the application and implication about this index should be careful and tested further.

Author Response

1.     Reviewer 3

Comments and Suggestions for Authors

This is an interesting study. Authors want to use a integrated index to assess the risk of infant food security. This simple index might be useful or implication for child health care. However, the following questions should be notes.

1. In the part of methods, the concept of index establishment should be presented in detail. How many subjects were used for setting up index and how many subjects for verification?

Response:  We have added the following statements to address the issue of how many subjects were used to set up the index (Ln125-126): “After excluding subjects for whom data on food frequency, breastfeeding, and coping was incomplete or missing, the final sample size for the food security index was 6,467 infants.”

We have provided a Consort Diagram to support this.

We have added the following statement as a footnote in Table 6 to clarify the number of subjects used in verification:

g If n<6467, incomplete and missing data has reduced the sample size.

2. Authors mentioned the CFA was used for establishment of index. However, there was nothing to say about this CFA. If CFA was used, the theoretical concept about this index should be reported and the procedure to test it by CFA was also reported.

Response: We used CFA as an initial guide to variable selection for the index, but the final variables included in the food security index were based on the capacity of the data to inform on food security, and literature review.  Given that CFA was not the sole method for index development, we have removed reference to CFA.

We have also removed reference to CFA from the abstract (Ln17-18).  The original sentence was: “A FS index was derived using confirmatory factor analysis.  Associations were assessed by logistic regressions and described…”  The revised sentence is: “A FS index was derived, and associations were assessed by logistic regressions and described…”

3. The calculated index FIS was normal distribution? or what kind of distribution for it? this problem is unclear.

Response:  We have added the following statement (Ln 172-73): The calculated food security index is normally distributed (x ̅=25.52,s.d.=5.07).

4. For table 2, the weights were unclear. what is criteria for the weights? Some statement might be necessary. And the application and implication about this index should be careful and tested further.

Response:  We have added the following statement (Ln129-134) to address this point: “In order to combine simplicity in assessing diet quality with the ability to differentiate healthy foods from unhealthy ones [33], punitive scoring is applied to unhealthy foods and use of coping methods.  Positive points were awarded for the consumption of healthy foods and breastfeeding.  Scoring was reversed and points deducted for consumption of unhealthy food and the use of coping methods.  A zero-score was awarded to non-breastfeeding and not using coping methods. ”

Round 2

Reviewer 3 Report

The authors have addressed most of my comments. And the manuscript has been improved much. The following points should be improved further.

About CFA, some statement like " ....our final choice of breastfeeding variable, as determined by CFA,.... " still be presented. If CFA is mentioned, some statments about this methods should be provided in the part of methods.(L112-113)

Authors said they think the distribution of the index is normal, bu no statistical test for normality was presented but just provided the mean and sd. Because the cut-off points were based on mean and sd (Table 8), which is suitable to describe the normal distrituion.  pls think about this problem.

Author Response

1.     Reviewer 3 – second round of comments

Thank you for your detailed and considered review of this paper.  We are pleased that we have been able to address most of your comments, and will attempt to address the remaining two points with this response.

1. About CFA, some statement like " ....our final choice of breastfeeding variable, as determined by CFA,.... " still be presented. If CFA is mentioned, some statments about this methods should be provided in the part of methods.(L112-113)

Response:  As you suggest, we have restored discussion of CFA to the paper.  We refer to this first in the abstract: “A FS index was derived using confirmatory factor analysis.  ” (Ln17).  Then, we have added the following paragraph:

“We treated infant FS as a latent construct, and used confirmatory factor analysis (CFA) to assess the extent to which observed data agrees with theoretical concepts of FS.  A four-factor model was adopted.  The choice of the indicators has been informed by the literature, the aim of study, and the availability of data with sufficient coverage.  In the proposed model, it was hypothesized that 19 items measure four constructs of food security; sentinel food group consumption (selected staple or nutrient-dense foods), EDNP consumption, maternal coping methods, and breastfeeding. Properties of the model were studied using CFA as implemented in AMOS 25.0.[30] The analysis produces statistics reflecting overall fit of the observed data to the model, together with estimates of correlations between the items and factors.  This modified model (15 items) for the overall sample showed adequate model fit, χ2 = 266.503, p < 0.001, RMSEA= 0.03, CFI = 0.94.” (Ln 95-104)

2. Authors said they think the distribution of the index is normal, bu no statistical test for normality was presented but just provided the mean and sd. Because the cut-off points were based on mean and sd (Table 8), which is suitable to describe the normal distrituion. pls think about this problem.

Response:  We have added the following statement (Ln 183-185): “…with skewness of -0.004 and kurtosis of 0.397.  We assess normality using skewness and kurtosis of the distribution, because this method is may be relatively correct in both small samples and large samples.[36] 

We have provided an additional reference to support this statement: Kim, H.-Y. Statistical notes for clinical researchers: assessing normal distribution (2) using skewness and kurtosis. Restorative dentistry & endodontics 2013, 38, 52-54, doi:10.5395/rde.2013.38.1.52.